# EXPLOITING STRUCTURED DATA FOR LEARNING CONTAGIOUS DISEASES UNDER INCOMPLETE TESTING

## ABSTRACT

One of the ways that machine learning algorithms can help control the spread of an infectious disease is by building models that predict who is likely to get infected making them good candidates for preemptive interventions. In this work we ask: can we build reliable infection prediction models when the observed data is collected under limited, and biased testing that prioritizes testing symptomatic individuals? Our analysis suggests that when the infection is highly contagious, incomplete testing might be sufficient to achieve good out-of-sample prediction error. Guided by this insight, we develop an algorithm that predicts infections, and show that it outperforms baselines on simulated data. We apply our model to data from a large hospital to predict Clostridioides difficile infections; a communicable disease that is characterized by asymptomatic (i.e., untested) carriers. Using a proxy instead of the unobserved untested-infected state, we show that our model outperforms benchmarks in predicting infections.

## 1 INTRODUCTION

Preemptively identifying individuals at a high risk of contracting a contagious infection is important for guiding treatment decisions to mitigate symptoms, and preventing further spread of the contagion. In this paper, we study how to build individual-level predictive models for contagious infections while explicitly addressing the challenges inherent to contagious diseases.

Building accurate infection prediction models is hindered by two main factors. First, contagious infections defy the usual *iid* assumption central to most machine learning methods. This is because an individual's infection state is not independent of their contacts' infection states. Previous work has often relied on expert knowledge to construct exposure proxies (Wiens et al., 2012; Oh et al., 2018). It is then assumed that conditional on the exposure proxy and individual characteristics, individual outcomes are independent of one another. Second, the observed data is biased due to incomplete testing. We use the term "incomplete testing" to describe the scenario where only a small, biased subset of infected individuals get tested. Such a scenario is ubiquitous in the context of contagious infections for several reasons. While many individuals carry the pathogen, only a fraction display symptoms. Even in the presence of unlimited testing resources, the latter are far more likely to get tested leading to biased data collection where individuals predisposed to displaying symptoms are over-represented. Incomplete testing makes learning accurate models difficult since the collected labels are missing not at random leading to biased, inconsistent estimates.

In this work, we treat non-independence of outcomes as a blessing rather than a curse. Our proposed approach leverages the fact that an individual's infection state provides useful information about their contacts' true infection states. This information is used to generate pseudo-labels for untested individuals, mitigating issues due to incomplete testing. The key idea behind our approach is that highly structured patterns of contagion transmission can serve as a complementary signal to identify even untested carriers. The stronger that signal is, the less impact that incomplete testing will have. Our contributions can be summarized as follows: (1) We identify two properties of the collected data that can be exploited to mitigate the effects of incomplete testing. (2) We propose an algorithm that leverages that insight to predict the probability of an untested individual carrying the disease. (3) We empirically evaluate the effectiveness of our method on both simulated data and real data for a common healthcare associated infection. We show that predictions from our model can be used to

inform efficient testing and isolation policies. Using real data, we show that our model outperforms baselines in the task of predicting a hospital associated infection.

## 2 RELATED WORK

**Infectious disease modeling.** Modeling the transmission of infectious diseases has been extensively studied in the epidemiology literature using SIS/SIR models and several other variants (Kermack & McKendrick, 1927). These epidemiological models focus on the *aggregate* levels of infections in a community, which is distinct from our approach here where we focus on predicting individual level infections. In the machine learning literature, previous work has relied on proxies for exposure, e.g., the prevalence of a disease in a community (Wiens et al., 2012; Oh et al., 2018), and implicitly assume that conditioning on individual characteristics. Similar to our approach, Fan et al. (2016) and Makar et al. (2018) take into account structured data, namely contact networks to compute infection estimates (Fan et al., 2016; Makar et al., 2018). We differ from these approachs in that (1) we do not make parametric assumptions about the joint distribution of the observed or latent variables, and instead use nonparametric models (neural networks) to model the infection states, (2) we do not assume all infections will become symptomatic as is done in Fan et al. (2016), and (3) unlike the approach taken by Makar et al. (2018), we model time evolving sequences of infections taking into account the exposure states of potential asymptomatic carriers.

**Semi-supervised learning.** Our proposed approach relies on transductive reasoning to generate labels for untested individuals. In that, it is closely related to semi-supervised learning methods, such as pseudo-labeling (Lee, 2003), and self-training (Robinson et al., 2020). However, in traditional pseudo-labeling, the transductive power comes from the fact that points similar to each other in the input space should have similar outputs. Here, the rich structure in the data allows for more: we can construct pseudo-labels for untested individuals not just by relying on their similarity to other labeled instances, but also by observing their observed contacts' infection states. Our empirical results, and analysis are similar in spirit to concepts presented in the semi-supervised literature, specifically the cluster assumption, which we discuss at length later (Seeger, 2000; Rigollet, 2007).

**Graph Neural Networks.** Our proposed approach incorporates knowledge of the contact network. In that it is similar to Graph Neural Networks (GNNs), which utilize relational data to generate prediction estimates (Zhou et al., 2018). GNNs fall into two categories, the first relies on transductive reasoning and cannot generalize to new communities (e.g., Kipf & Welling (2017)) or inductive, which can be used to generate estimates for previously unseen graphs (e.g., Hamilton et al. (2017)). Our work is similar to the latter category with an important distinction: our approach leverages unlabeled data giving more accurate, and robust estimates.

Our work can be viewed as combining the strengths of semi-supervised learning, and GNNs to address limited testing. In addition, our approach augments the strengths of those two approaches with ideas from domain shift, and causal inference such as importance weighting (Cortes et al., 2010) to address biased testing.

## 3 PROBLEM SETTING

**Setup.** Let $y^t \in \{0, 1\}$ denote an individual's true infection state at time $t$, with $y^t = 0$ if an individual is not infected and 1 if they are. We use $\bar{\mathbf{x}}^t \in \mathcal{X}^t$ to denote a vector of the individual's features at time $t$, and define $J_i^t$ to be the set of indices of $i$'s contacts at time $t$. We assume that contact indices are known, i.e., that the contact network is observed. Let $e_i^t \in \mathbb{R}_{\geq 0}$ denote $i$'s exposure state, with $e_i^t = \sum_{j \in J_i^t} y_j^t$. The exposure state is fully observed only when all of $i$'s contacts have been tested, but otherwise either partially observed or unobserved. Define $\mathbf{x}^t = \bar{\mathbf{x}}^t || e^t$, where $||$ as the concatenation operator, i.e., $\mathbf{x}^t \in \mathcal{X}^t \times \mathbb{R}_{\geq 0}$. Let $o^t \in \{0, 1\}$ denote the observation state, with $o^t = 1$ if an individual's label is observed, i.e., if the individual has been tested for the infection. We use the super-script $: t$ to denote variables from time $t = 0$ up to and including $t$, e.g., $\mathbf{x}^{:t} = [\mathbf{x}^0, ..., \mathbf{x}^s, ..., \mathbf{x}^t]$.

Throughout, we use capital letters to denote variables, and small letters to denote their values. We use $P(\boldsymbol{X}^t, O^t, Y^{t+1})$ to denote the unknown distribution over the full joint. Under biased testing, we have that $P(\boldsymbol{X}^t | O^t = 1) \neq P(\boldsymbol{X}^t | O^t = 0) \neq P(\boldsymbol{X}^t)$. We assume that $0 < P(O^t = o | \boldsymbol{X}^t = \mathbf{x}) < 1$, for all $\mathbf{x} \in \mathcal{X}$, and $o \in \{0, 1\}$. This is the same as the overlap assumption

in causality literature. In addition, we assume that $i$'s outcome is independent of their contacts given $\mathbf{x}_i$, which is itself a function of the contacts' outcomes, we refer to this as the conditional independence assumption. We consider the case where we have access to (1) a labeled (i.e., tested) set of individuals $\mathcal{D}_1 = \{\mathcal{D}_1^t\}_{t=0}^T = \{(\mathbf{x}_i^t, y_i^t), \dots (\mathbf{x}_{n_1^t}^t, y_{n_1^t}^t)\} \sim P(\boldsymbol{X}^t, Y^{t+1} | O^t = 1)$, and (2) an unlabeled (untested) set of individuals $\mathcal{D}_0 = \{\mathcal{D}_0^t\}_{t=0}^T = \{\mathbf{x}_i^t, \dots, \mathbf{x}_{n_0^t}^t\} \sim P(\boldsymbol{X}^t | O^t = 0)$, such that for each $i \in \mathcal{D}_0 \cup \mathcal{D}_1$, and each $t \in [0, T]$, we have that $J_i^t \in \mathcal{D}_0 \cup \mathcal{D}_1$. It will also be convenient to use $\mathcal{U}^t$ to denote the set of indices of untested individuals at time $t$.

**Learning objective.** We are interested in learning $f : \mathbf{x}^{:T} \to y^{T+1}$. To focus the discussion on the novel component of our approach, we consider a setting where we are only interested in predicting the outcomes for a single time step. It will be particularly useful to consider the task of making predictions for $t = 2$, using data from $t = 0, 1$, dropping the time superscript when it can be inferred from the context. We present the full model predicting infection sequences over time in section 5. Let $\ell$ be the logistic loss function. Our goal is to find $f \in \mathcal{F}$, where $\mathcal{F}$ is some hypothesis space such that the risk of incorrectly classifying the infection state $\mathcal{R}_f = \mathbb{E}_{\boldsymbol{X}, Y}[\ell(f(\boldsymbol{X}^t), Y^{t+1})]$ is minimized. We briefly consider a scenario where we have oracle access to the true exposure states but we return to the more realistic, non-oracle scenario later. Under the conditional independence assumption, we can break down the risk to the sum of independent losses. Define the inverse probability of being tested, $w^t(\boldsymbol{X}) = P(O^t = o)/P(O^t = o | \boldsymbol{X}^t)$, following Robins (1998), and Robins et al. (2000). Due to the overlap assumption, and under biased testing, we have that:

$$\mathcal{R}_f = \mathcal{R}_f^{w^t} = \mathbb{E}_{\boldsymbol{X}, Y | O=1}[w^t(\boldsymbol{X})\ell(f(\boldsymbol{X}), Y) | O = 1], \tag{1}$$

(Cortes et al., 2010). The reweighted risk simply places a higher importance on the loss of individuals who are unlikely to be tested. $\mathcal{R}_f^{w^t}$ cannot be directly computed since the expectation is defined with respect to the unobserved distribution. However, the following reweighted empirical loss is an unbiased estimator of $\mathcal{R}_f^{w^t}$:

$$\varepsilon(f) = \sum_{i \in \mathcal{D}_1^t} w_i^t \ell(f(\mathbf{x}_i^t), y_i^{t+1}),$$

by Cortes et al. (2008), where $w_i^t = p(O^t = o_i^t)/g(o_i^t | \mathbf{x}_i^t)$, $p(O^t = o_i^t)$ is the empirical estimate of $P(O^t = o)$, and $g(o_i^t | \mathbf{x}_i^t)$ is the estimated probability of getting tested conditional on individual characteristics. Without oracle access to exposure states, the samples $\mathbf{x}^t \sim P(\boldsymbol{X}^t | O^t = 1)$ are incomplete. This is because $\mathbf{x}_i^t$ includes $e_i^t$, which is a function of $y_j^t : j \in J_i^t$. We only fully observe $e_i^t$, and hence $\mathbf{x}_i^t$ for individuals whose contacts have all been tested. To address this, we define $Q(\mathcal{D}_1^t)$, a set of partially imputed distributions that are consistent with the labeled samples. It is the set of all possible distributions over the (partially) unobserved $e_i^t$. Our risk is now defined with respect to both $Q$, and $f$, and our task is to find $Q$ and $f$, such that the following empirical risk is minimized:

$$\varepsilon(f, Q) = \sum_{i \in \mathcal{D}_1} \hat{w}_i^t \ell(f(\hat{\mathbf{x}}_i^t), y_i^{t+1}), \tag{2}$$

where $\hat{\mathbf{x}} = \bar{\mathbf{x}}^t || \hat{e}^t$, and $\hat{e}_i^t \sim Q$, and $\hat{w}_i^t = p(O = o_i)/g(\hat{\mathbf{x}}_i, o_i)$. Minimizing this objective is prone to extreme overfitting. To see why, consider some $Q$ that sets $e_i^t = 100$ for every $i : o_i^t = 1, y_i^{t+1} = 1$, and 0 for $i : o_i^t = 1, y_i^{t+1} = 0$. Since $Q$ is essentially leaking the true label into the input space, it is trivial to find some $f$ that takes in the imputed inputs, $\{(\bar{\mathbf{x}}_i^t, 100, y_i^{t+1})\}_{i:y_i^{t+1}=1}$, and $\{(\bar{\mathbf{x}}_i^t, 0, y_i^{t+1})\}_{i:y_i^{t+1}=0}$ and gives perfect performance. Such an $f$ is clearly expected to have poor generalization error. We next consider how to leverage existing properties of the problem as efficient regularizers.

## 4 Exploiting structure as a regularizer

We seek to constrain the candidate sets $\mathcal{F}$ and $Q(\mathcal{D}_1)$ to ensure that such pathological overfitting is avoided. To do so, we exploit the structure in the data, namely the interdependence among individuals' infection states, and the availability of unlabeled data. Recall that the exposure state is the sum of the contacts' infection states. This means that when we draw $\hat{e}_i^t$ from $Q$, *we are implicitly drawing a label for $i$'s contacts' outcomes, by definition of $\hat{e}_i^t$.* This becomes obvious if we breakdown $\hat{e}_i^t$

draws from $Q$ as follows: $\hat{e}_i^t = \sum_{j \in J^t(i)} \mathbb{1}\{j : o_j^t = 1\} \cdot y_j^t + \mathbb{1}\{j : o_j^t = 0\} \cdot \hat{y}_{i,j}^t$, and $\hat{y}_{i,j}^t \sim Q$, such that $\hat{y}_{i,j}^t \in [0,1]$. This breakdown immediately implies two properties that should hold for "good" $Q$'s. First, $Q$ should assign the same estimate for the same individual. Consider the case where two $i = a$, $i = b$ come into contact with the same individual $j$, who has not been tested. For simplicity, suppose that $a$, and $b$ have no other contacts. In this case, one failure mode would be if $Q$ assigns $e_a^t = \hat{y}_{a,j}^t = 0.9$, and $e_b^t = \hat{y}_{b,j}^t = 0.1$, which constitutes two "votes" on the true state of $j$; one vote by each contact. Second, note that $Q$ is implicitly assigning pseudo-labels for the infection states of untested contacts, this means that $Q$'s imputed labels should be similar to the labels predicted by $f$. Going back to the previous example, besides having $\hat{y}_{a,j}^t = 0.9$, $\hat{y}_{b,j}^t = 0.1$, we also have $f(\mathbf{x}_j^t) = 0.4$. A good regularization method should then explicitly encourage the pseudo-labels to be similar to the estimated labels from $f$, and hence implicitly penalizing varying estimates for the same individual. This intuition is encoded in the main loss in our proposed approach:

$$f^*, Q^* = \min_{f,Q} \frac{1}{n_1^t} \sum_{i:o_i^t=1} \hat{w}_i^t \ell(f(\hat{\mathbf{x}}_i^t), y_i^{t+1}) + \frac{\lambda}{|J_i^t \cap \mathcal{U}^t|} \sum_{j \in J_i^t \cap \mathcal{U}^t} \hat{w}_j^{t-1} \ell(\mathbb{1}\{f(\mathbf{x}_j^{t-1}) > \tau\}, \hat{y}_{i,j}^t) \quad (3)$$

where $|.|$ denotes the set cardinality, $\lambda \geq 0$, and $\tau$ are parameters to be picked via cross validation, $\hat{y}_{i,j}^t \sim Q^*$, and $\hat{e}_i^t = \sum_{j \in J^t(i)} \mathbb{1}\{j : o_j^t = 1\} \cdot y_j^t + \mathbb{1}\{j : o_j^t = 0\} \cdot \hat{y}_{i,j}^t$. When $\lambda > 0$, this objective is somewhat similar to pseudo-labeling (Lee, 2003), it would encourage the votes of each of $j$'s contacts to conform with the prediction from $f$, and implicitly with one another. When $\lambda = 0$, equation 3 prioritizes finding good predictions for the labeled data, ignoring possible structure implied by the data. Note that in the second term in equation 3, we have $f(\mathbf{x}_j^{t-1})$, $f(\hat{\mathbf{x}}_j^{t-1})$, meaning we assume not imputed exposure component for contacts at time $t-1$. This is only because we are considering the simple setting where we make predictions for $t+1 = 2$, meaning at $t-1 = 0$, this is the beginning of the observation period and we cannot impute exposure yet. We consider more complicated settings where the contacts' inputs also include an exposure state later.

## 4.1 WHEN DOES STRUCTURE WORK AS A REGULARIZER?

We now ask: when do we expect equation 3 to yield models superior to those that ignore structure? First, if the imputed $\hat{y}_{.,j}$ concentrates around significantly different values for $j : y_j = 1$, and $j : y_j = 0$, then we expect minimizing equation 3 to yield better models. We stress that we do not require $\hat{y}$ to be an accurate estimate of the true labels, but only require that there is significant *separation* between the imputed values for untested-infected individuals and untested-uninfected individuals, i.e., they are distinguishable. This distinction means that even noisy and inaccurate estimates of $\hat{y}$ can be sufficient. We expect such high separability to exist, even in settings of low and biased testing if observed data satisfies a property which we will refer to as carrier potency. The carrier potency property can be viewed as an extension of the margin condition in classification (Tsybakov et al., 2004; Audibert et al., 2007): it states that there are few $j$'s who have contacts with ambiguous infection states. In other words, infections cluster so that infected-untested individuals tend to have many more infected contacts than do uninfected-untested. Such a condition will be satisfied if the infection is highly contagious. We refer to this property as the **potency property**.

Second, even if the imputed $\hat{y}$ allows high seperability, but $\hat{x}$ makes it difficult to identify a learnable mapping from $\hat{x}$ to $\hat{y}$, then minimizing equation 3 instead of the objective on the labeled data only does not help. Such is the case when untested-healthy and untested-infected individuals "look" the same, meaning they have very similar characteristics and exposure states. This property is often referred to as the cluster assumption in semi-supervised learning literature (Rigollet, 2007; Seeger, 2000). The cluster assumption states that individual characteristics, and exposure states tend to form near discrete clusters, with homogeneous labels within each cluster. We refer to this property as the **dissimilarity property**.

The degree to which these two properties are satisfied in the observed data will depend on the infection being studied, the environment in which it is spreading, and the quality of data collection, among others. Importantly, as we show empirically in section 6, our proposed approach "does no harm" in that in the worst case scenario, when these two properties degrade to the point of non-existence, our model performs as well as the best baseline.

## 5 PROPOSED METHOD

Our proposed model, a Model for Infections under Incomplete Testing (MIINT) leverages labeled and unlabeled data in order to predict sequences of infections over time. MIINT minimizes a slight variant of equation 3, which is modified to predict sequences of infections. Let $\mathcal{A}_i^t$, be the set of ancestors of $i$ at time $t$ whose outcomes are unobserved, i.e., $\mathcal{A}_i^t = J^t(i) \cap \mathcal{U}^t$, $\mathcal{A}^{t-1} = \bigcup_{j \in \mathcal{A}_i^t} J^{t-1}(j) \cap \mathcal{U}^{t-1}$, etc. The loss at time $t$ is defined as:

$$\mathcal{L}^t = \frac{1}{n_1^t} \sum_{i \in \mathcal{D}_1} \hat{w}_i^t \ell(f(\hat{\mathbf{x}}_i^t), y_i^{t+1}) + \sum_{s=0}^t \frac{\lambda}{|\mathcal{A}_i^t|} \sum_{j \in \mathcal{A}_i^s} \hat{w}_j^s \ell(\mathbb{1}\{f(\hat{\mathbf{x}}_j^s) > \tau\}, \hat{y}_{i,j}^s), \quad (4)$$

and the final objective is to find $f^*, Q^*$, such that: $f^*, Q^* = \min_{f,Q} T^{-1} \sum_t \mathcal{L}^t$. We assume that $f$ does not vary over time (though that is an assumption that could be relaxed), and take $\mathcal{F}$ to be the space of recurrent neural networks (RNNs), which are ideal for modeling sequences of data. We propagate the predicted state forward in time, meaning $f$ takes in $\mathbf{x}^t, e^t$ and $\hat{y}^t$ to predict $\hat{y}^{t+1}$. This ensures that exposures at time $< t$ are taken into account when predicting at time $t$. Note that equation 4 can be broken down into the independent sums of individual losses, as well as their ancestors' losses. This means we can use stochastic gradient descent, with gradient updates defined with respect to mini-batches as is typically done. One limitation is that equation 4 as stated would require keeping track of all the ancestors' states since $t = 0$, which can be prohibitive. For long observation periods, we suggest considering a subset of $\mathcal{A}_i^t$ up to a reasonable time limit.

The algorithm used to train MIINT, similar to pseudo-labeling (Lee, 2003), is inherently an expectation maximization algorithm, where we iterate between computing the expected label for the untested samples (i.e., finding the optimal $\hat{Q}$, and identifying the optimal of $f$ that maximize the likelihood of the observed labels under $\hat{Q}$) until convergence. Convergence is achieved when the change in loss defined over the samples with observed labels in a held out validation set $< \epsilon$ for some small $\epsilon$. For our purposes, we find it sufficient to let $Q$ be a deterministic function rather than an actual distribution. However, our approach is trivially extendable to allow $Q$ to be a distribution, for example using techniques described in Tran et al. (2017). All models presented in this paper are implemented using Tensorflow (Abadi et al., 2016).

Finally, recall that we need to estimate $\hat{w}_i^t = \frac{p(O = o_i)}{g(\hat{\mathbf{x}}_i, o_i)}$. We follow Chernozhukov et al. (2017) in using an independent sample to estimate $g$. Importantly, $g$ depends on $\hat{\mathbf{x}}$. So we follow an iterative process: after every epoch of training, we use the most updated $f$ to estimate the unobserved outcomes in the validation set, and hence get an estimate for $\hat{e}$ and $\hat{\mathbf{x}}$ for the independent weighting sample. We use these imputed values to learn an updated $g$. The updated $g$ provides estimates for the weights of the training samples of the main prediction model, which are used to reweight the loss function for the next epoch, and so forth.

## 6 EXPERIMENTS

We evaluate our model on a simulated and a real data setting. In the simulated setting, unlike the real data setting, we have access to the true infection state, which allows us to evaluate the performance of the model and baselines under different patterns of infection. In both settings, we present results from our model (MIINT) and four baselines: (1) Optimistic Model (**OM**): a model that assumes that all unobserved labels are $= 0$, (2) No Exposure Model (**NEM**), a model that ignores exposure, and attempts to predict infections solely based on the individual characteristics, (3) GraphSAGE (**GNN**) a graph neural network that takes into account the contact network, and observed infection states (Hamilton et al., 2017) but ignores untested individuals, and (4) Pseudo-Labeling (**PL**) a semi-supervised learning method that takes into account untested individuals but ignores the rich graph structure (Lee, 2003).

For all models, we weight the loss from each individual by the inverse of their estimated propensity to be tested, $w_i^t$, which is estimated using an independent sample following Chernozhukov et al. (2017). For our model, we use the iterative weighting technique outlined in section 5. We also present an ORacle Model (**ORM**): an unrealistic model that has oracle access to the true labels for the whole population. For all these models, we keep the neural network architecture fixed. We use cross-validate to get the values of $\lambda, \tau$. Results from unweighted models, and details about cross-validation and network architecture are included in the supplement.

## 6.1 SIMULATION EXPERIMENTS

Our goal here is to highlight how MIINT can be used to inform testing and isolation policies that lead to reduction in infection rates, as well as empirically validate our conjectures regarding favorable properties under which MIINT is expected to be superior.

**Setup.** We simulate a world where there are three types of people: symptomatic if exposed ($G_0$), asymptomatic if exposed ($G_1$), and immune ($G_2$). If exposed, individuals in group $G_0$ become infected and symptomatic, hence they are more likely to get tested. If exposed individuals in group $G_1$ become infected without displaying symptoms. This group is unlikely to get tested. Finally, $G_2$, the immune group is unlikely to get the infection even if exposed. To simulate individuals' characteristics (i.e., $\overline{\mathbf{x}}$), we map the distinct groups to distinct MNIST digits. We use MNIST images because (1) they provide a complex input space compared to randomly generated data, and (2) images can be easily classified as similar or dissimilar, which enables us to design experiments where the dissimilarity property can be easily manipulated as described later.

Let $\nu_i$ denote the pixels of an MNIST image $i$. For $G_0$ we randomly sample without replacement $n/3 \cdot T$ elements from the set $\{\nu_i\}_{i:d_i=0}$, where $n$ is the total sample size. For $G_1$, and $G_2$ we sample from $\{\nu_i\}_{i:d_i=1}$, and $\{\nu_i\}_{i:d_i=2}$, respectively. Note that the infection states will be different within each group, since infection also depends on the exposure state, and injected noise. We draw the edge sets $\{J^t(i)\}_{i \in n, t \in [0,T]}$ according to a stochastic block model, parameterized by the matrix $B$, where $B_{k,l}$ is the probability that an individual from $G_k$ forms an edge with an individual from $G_l$. $B$ is important in simulating different levels of carrier potency. When $B_{1,k}/B_{1,2}$ for $k = \{0,1\}$ approaches 1, members of the asymptomatic carrier group is equally likely to form an edge with individuals who are infectable ($G_0$) as with individuals who are immune ($G_2$). This is the unfavorable low-separation setting. On the other hand, if $B_{1,k}/B_{1,2} = 5$, for example, individuals in $G_0$, and $G_1$ are 5 times more likely to form an edge with someone in a susceptible group as compared to forming an edge with an individual in $G_2$. This is a favorable, high-separation setting.

We mimic the situation where testing started after a significant proportion of the population has been exposed by randomly setting the true exposure state of 20% of the population to be 1 at time $t = 0$. Exposure for each individual $e_i^t = \sum_{j \in J_i^t} y_j^t \geq 1$. The true infection label $y_i^{t+1} = \mathbb{1}\{i \in (G_0, G_1)\} \cdot \mathbb{1}\{e_{i,t} = 1\}$. We introduce noise by randomly flipping the labels of 1% of the population. If an individual tests positive at $t < T$, their label remains positive until $t = T$. We define $p_{\text{obs}}$ to be the proportion tested, and hence their true label is observed. We pick the probability of observing $i$'s label based on his true infection state, meaning, $p(o_i|y_i = 1) \neq p(o_i|y_i = 0)$. For all the simulations, we set $T = 6$, we draw $500 \times 6$ samples for each of the training, validation, and testing sets. We simulate an independent sample to compute the weights $w_i$, so we also draw $500 \times 6$ samples that will be used to train and validate weighting models. For each experiment, we draw 10 different datasets, and report the mean and standard deviation of the performance metric across the 10 draws.

**Informing testing and isolation policies.** Here, we highlight how our model can inform efficient testing and isolation policies. We simulate biased and limited testing by setting $p(o_i|y_i = 1)/p(o_i|y_i = 0) = 5$, and $p_{\text{obs}} = .1$ respectively. We set $B_{1,k}/B_{1,2} = 5$, making it a high potency setting where MIINT is expected to perform well. We mimic a situation where no isolation interventions are taken at training time. At test time, we fix a testing budget of at most $p_{\text{test}}\%$ of the total population on each time step. We use the predictions from each model to inform who gets tested by picking the top $p_{\text{test}}\%$ with the highest predicted probability of infection. Of those tested, individuals who are truly infected are "isolated" by setting their edges for the subsequent time steps to 0. They are also taken out of the population eligi-

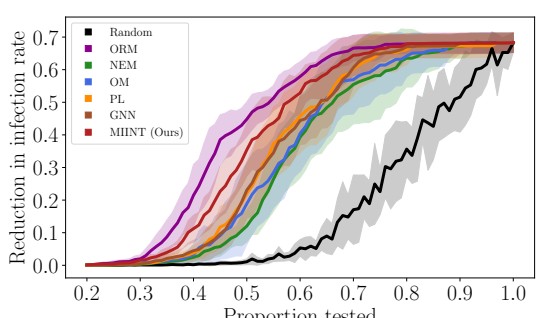

Figure 1: Reduction in infection rates relative to a policy that does not isolate infections (no-action policy) as the daily testing budget varies. Our model achieves the highest reductions in policy relative to all realistic (i.e., non-oracle) models.

ble for further testing. We compute the infection rate, $\pi_M$ for a model $M$ as $\pi_M = n^{-1} \cdot \sum_i \max_t y_{i,t}$. We define $\pi_0$ as the infection rate under a no-action policy, that is if no isolation interventions are taken. Our main metric of interest is the reduction in infection rate relative to the no-action policy $= \pi_0 - \pi_M/\pi_0$. Figure 1(left) shows the reduction in infection rate on the $y-$axis for different values of the testing budget $p_{\text{test}}\%$ on the $x-$axis. In addition to the main baselines, we also show results from a random testing policy. The results show that for any given testing budget, our model outperforms all feasible baselines giving higher reduction in infection rates. Importantly, the results imply that our model is able to achieve near oracle infection control with only $70\%$ testing, compared to $\approx 90\%$ for the baselines.

In next two settings, we empirically validate our conjectures about the two properties which enable our model to outperform others, and explore what happens as these favorable properties are weakened to the point of non-existence.

**Sensitivity to the potency property.** Here, we fix $p_{\text{obs}} = .1$, and $p(o_i|y_i = 1)/p(o_i|y_i = 0) = 5$ but sweep over carrier potency by varying the value of $B_{1,k}/B_{1,2}$ from 1 (low potency) to 5 (high potency). Figure 2(left) shows $B_{1,k}/B_{1,2}$ on the $x-$axis and the AUROC on the $y-$axis. The plot shows that MIINT outperforms other baselines when there is high potency, and as potency declines, its performance becomes similar to that of the other baselines. This supports our conjecture that our regularization approach is advantageous when it is easy to impute the true infection states for individuals based on their contacts' infection states.

**Sensitivity to the dissimilarity property.** Here we examine what happens when the cluster assumption breaks down, meaning when untested individuals with similar characteristics have different infection states. We do so by moving the untested, and possibly infected[1] individuals to "look" similar to the untested-healthy. Specifically, we sample pairs of images $\{(\nu_i, \nu_j)\}_{i,j:d_i=1,d_j=2}$. We then use VoxelMorph (Balakrishnan et al., 2018), a learning-based framework for deformable, pairwise image registration to learn a function that gives us a deformation field which we then apply it to pairs of images, moving $\nu_i$ to look closer to $\nu_j$. Further details about the deformation process are in the original VoxelMorph paper. Figure 2(right) shows the results of this setting. The $x-$axis can be viewed as the degree of similarity between the two untested groups with 0 being dissimilar (i.e., the original images without any deformation) and 1 being very similar (i.e., all images of the digit 1 look almost identical to 2's). The $y-$axis is the average AUROC. We see that all models perform worse as members in $G_1$ look more and more similar to those in $G_2$. We also see that MIINT outperforms all baselines when the two groups are dissimilar, but performs as well as the others when the mapping from input space to label becomes more difficult.

The last two experiments confirm our conjectures about the properties necessary for MIINT to perform well, and imply that MIINT "does no harm": at worst it performs comparably to alternatives, and at best it can give significantly better performance making it the superior alternative. Additional results examining the effect of bias and limited testing are in the supplement.

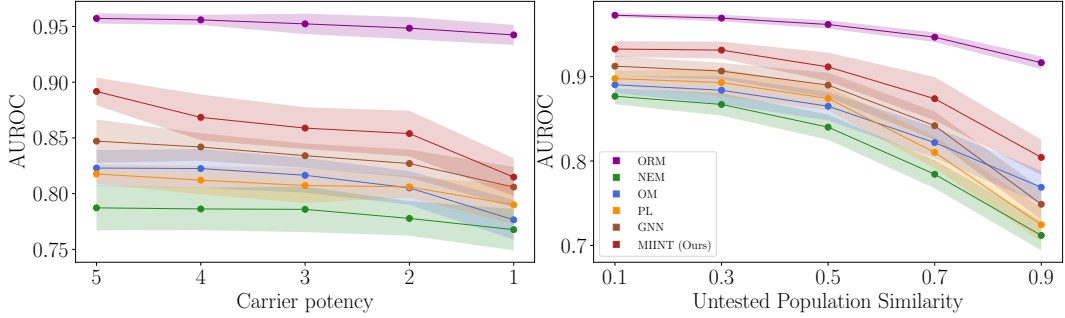

Figure 2: Left: Impact of varying levels of carrier potency controlled by $B_{1,k}/B_{1,2}$. Our model outperforms baselines, especially in cases with high potency. Right: Impact of high (=.9) and low (=.1) similarity between the characteristics of the untested-uninfected and untested-infected populations. Our model outperforms baselines when the two populations are dissimilar.

---

[1]Individuals in $G_1$ are only infected if they get exposed.

## 6.2 REAL DATA EXPERIMENT

Here, our task is to predict the onset of Clostridioides difficile infections (CDI) among patients in a large urban hospital. CDI is a contagious infection that attacks the gut, and causes over 300,000 infections annually in the US (Magill et al., 2014). As with most contagious infections, asymptomatic carriers of CDI exists and can contribute to the spread of the infection (Riggs et al., 2007).

**Setup.** Using Electronic Medical Records, we extract daily characteristics of patients who were admitted to the hospital between 09/01/2012 and 06/01/2014. We follow similar inclusion criteria as Oh et al. (2018); Makar et al. (2018), outlined in detail in the supplement. We collect all patient characteristics available upon admission (e.g., gender, age, medical history) as well as daily characteristics (e.g., lab tests). We collect contact networks, where an edge exists if two patients are in the same room on the same day or if they came into contact with the same nurse on the same day.

Here, we have partial access to the true infection states, since not all the patients are tested, making accurate evaluation of different models difficult. Therefore, we exploit testing protocols to construct a proxy "true" label and a proxy "observed" label. Whether a patient is labeled as CDI positive or not is a result of two, or possibly three tests. First, an enzyme immunoassay (EIA) and Glutamate dehydrogenase (GDH) test are conducted. If the results of the two tests are discordant, a polymerase chain reaction (PCR) assay acts as a tie-breaker. Previous studies comparing the outcomes of the two groups (those who have non-discordant EIA/GDH+ results vs. PCR+) have shown that the former experiences more severe complications (Origüen et al., 2018; Polage et al., 2015). This implies the EIA/GDH+ label can act a proxy for symptomatic infections, whereas PCR+ might be picking up on patients who are carrying the bacteria but have low toxin levels and therefore mild or no symptoms.

For this reason, we hide the PCR+ labels during training, presenting them as untested individuals to all models. At test time, we set the target label = 1 if a patient tested positive via EIA/GDH or PCR and 0 otherwise. In addition to the baselines outlined in section 6.1, we allow one of models full access to the EIA/GDH+ and PCR+ labels, and refer to it as a "partial oracle" model (POM) since it has access to the PCR+ labels, but not the full infection states. The latter are unavailable because the majority of patients are not tested. We also compare our results to the state-of-the-art prediction model for CDI, which is a logistic regression that takes into account the varying importance of different risk factors over the hospitalization, and relies on medical knowledge to construct exposure proxies. We refer to this model as the Expert driven Logistic Regression (ELR), details about the model are outlined in Wiens et al. (2012).

We split the data into 5 subsets based on time. The first subset holds 6 months of data and is used to train the main infection prediction models. The second and third subsets contain 5 months of data each, and are used for validation and testing of the main prediction model. The last 2 subsets are used for training and validation of the weighting models, and each contain 2 months worth of data. We report the AUROC, the True Positive Rate (TPR) at the threshold which achieves a False Positive Rate (FPR) of 10%, as well as the Area under the Precision Recall curve (AUPR) on the test set.

Table 1 shows the results of the models on the test set. For several models, the unweighted model outperforms its weighted counterpart. We show the better performing version here, and index it with "–U" to denote that it is the unweighted version. Results from all models, and results broken down by GDH/EIA+ vs. PCR+ are in the supplement. Standard deviations

|         | TPR@ FPR=10% | AUROC        | AUPRC        |
|---------|--------------|--------------|--------------|
| POM     | 0.49 (0.014) | 0.73 (0.003) | **0.2 (0.004)** |
| NEM–U   | 0.45 (0.009) | 0.7 (0.006)  | 0.13 (0.001) |
| OM–U    | 0.45 (0.012) | 0.7 (0.005)  | 0.12 (0.004) |
| ELR     | 0.53 (0.008) | 0.82 (0.006) | 0.09 (0.002) |
| GNN     | 0.24 (0.005) | 0.59 (0.005) | 0.03 (0.007) |
| PL–U    | 0.58 (0.012) | 0.78 (0.006) | 0.18 (0.009) |
| MIINT–U | **0.6 (0.007)** | **0.81 (0.006)** | 0.11 (0.002) |

Table 1: Performance metrics for CDI prediction on the test set.

are calculated by taking 100 bootstrap replicates of the test set data. We see that MIINT outperforms all others on all reported metrics. The one exception is ELR: MIINT and ELR achieve comparable AUROCs but MIINT significantly outperforms ELR on all other metrics. Unsurprisingly, MIINT outperforms POM even though the latter has access to better labels. We hypothesize that this is because in addition to accurately estimating the PCR+ patients, it is also capturing truly untested in-

fections, and utilizing these estimates to accurately impute the exposures of the EIA/GDH+ patients as well as the PCR+ patients leading to better performance metrics.

While POM, OM-U, NEM-U, and PL-U achieve higher AUPRC than our model, figure 8 in the supplement shows that these models achieve a higher AUPRC than MIINT because they have high precision at low recall, whereas MIINT has the highest precision at high recall values (greater than 0.5). High precision at low recall is not necessarily useful in the infectious disease setting, where having a low recall (meaning being unable to identify the true infections) can be disastrous.

## 7 CONCLUSION

We presented MIINT, a model that predicts contagious infections under biased and limited testing. It does so by taking into account contact networks and the interdependence of individuals' outcomes. We identified two properties that determine the extent to which MIINT outperforms other approaches. The first states that the more contagious the infection, the better MIINT performs. The second is the degree to which characteristics of untested and infected individuals and characteristics of the untested and healthy individuals form discrete clusters–an important property in general for semi-supervised learning. Using simulated data, we showed that MIINT can be used to guide testing policies that lead to reduced infection rates, and that even if the two properties outlined above are non-existent, MIINT still performs well. We showed that MIINT outperforms baselines when applied to real EMR data.

Because of the obvious relevance of our work to the current pandemic we should note that our model is best-suited for infections which spread within clinical environments (e.g., hospitals and nursing facilities). Within clinical settings, it is easier to track a individuals' contacts, which is necessary for our model. While our model can theoretically be used to predict COVID-19 infections in the community, it would require unprecedented levels of contact tracing, and data collection. In conclusion, we believe this work is a first step down an important path. If predictive models are to play a useful role in limiting the spread of contagious infections, they must take into account the interdependence of outcomes, and the fact that untested individuals are capable of spreading the disease before they have been diagnosed.

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
