# OpenReview forum: "Exploiting structured data for learning contagious diseases under incomplete testing"
_ICLR.cc/2021/Conference — Reject_

### Official Review · AnonReviewer4 · 2020-10-27
**A paper lacks necessary technical details and comprehensive comparisons**

**Rating:** 3
**Confidence:** 4

**Review:**

This paper focuses on contagious disease prediction with the consideration of observed data bias and patient exposure. The authors present a Model for Infections under Incomplete Testing (MIINT) and the experimental results show the proposed model outperforms baselines on *some* metrics.

Problems:
1. The paper is not well-written and pretty hard to follow.
- It is unclear why the authors use w^1(X) in eq (1). Why should t be set as 1?
- It would be better if the authors gave the definition of \hat{w}^t_i in eq (2).
- Q(D^t_1) is a set of imputed distribution. It is unclear how to generate Q.
- It would be better if the authors use consistent notations in different places. such as f(X,Y) and f(X^t, Y^{t+1})

2. The authors want to consider the exposure relationships between patients. An intuitive solution is graph-based models. It would be better if the authors compared the proposed model with graph-based models, e.g. [1]. Moreover, exposure states are not fully considered. The authors just simply use the count of exposure of observed true infection patients at the last time point. The exposure before two time-points is ignored.

[1] Xiaojin Zhu, Zoubin Ghahramani, and John D. Lafferty. "Semi-supervised learning using gaussian fields and harmonic functions." Proceedings of the 20th International Conference on Machine learning (ICML-03). 2003.

3. In Table 1, the two simple baselines (OM, NEM) and POM outperform the proposed model a lot on AUPRC. It is worth explaining why.

4. Due to missing technical details (especially how to generate Q), it is hard to re-implement the proposed models. It is necessary to provide code and data as supplementary materials.

---

> ### Author Response · Authors · 2020-11-25
> **Additional comparisons, and clarifying implementation details**
>
> 1a. We thank the reviewer for catching our typo. Equation 1 should have read w^t(X) not w^1(X). We have fixed that typo.
>
> 1b. We had originally included the definition of \hat{w} in section 5. We agree that this definition should have been included earlier as the reviewer points out, so we have updated the paper accordingly.
>
> 1c. Section 5 outlines how to generate this distribution. Specifically, below equation 4, we explain that the optimal Q is the minimizer of equation 4, and in paragraph 2 in section 5, we explain that Q is a deterministic function, but extensions of our approach to probabilistic functions are possible. Q can be thought of as the imputed infection states that maximize the likelihood of the observed data.
>
> 1d. We stress that f is never a function of X, and Y. Only the loss \ell, is a function of f(x) and y^{t+1}.
>
> 2. The paper that the author points to falls under the category of semi-supervised learning. We have implemented pseudo labeling (PL)  a neural network based semi-supervised learning model (See Lee 2003). This sets up an “apples to apples” comparison since our suggested approach is also based on a neural network. Our original conclusions hold even with the additional benchmark. In addition, our model does not simply count the number of observed infections in the last time points: (1) it sums the observed infections as well as the predicted infections in the previous time point, (2) our RNN model propagates the hidden patient state over time, which in turn propagates exposures at previous time points. We have explicitly stated that property in section 5 paragraph 1. We also stress that all the baselines have a similar network structure, so they also propagate the hidden states, along with their exposure proxies to future states.
>
> 3. As noted in the general comment: we found that models which achieve higher AUPR do so because they have high precision at low recall, whereas our model has the highest precision at high recall values.  High precision at low recall is not necessarily useful in the infectious disease setting, where having a low recall (meaning being unable to identify the true infections) can be disastrous. We added the Precision recall plot in the supplement.
>
> 4. We have also included our main model code in the supplement for re-implementation

---

### Official Review · AnonReviewer3 · 2020-10-28
**Interesting Paper; Some Confusions**

**Rating:** 4
**Confidence:** 2

**Review:**

Interesting Paper; Some Confusions

This paper considers the problem of predicting the infection status of untested individuals. The authors propose a recurrent neural network-based model to impute the infection statuses over time, incorporating information on both the individuals (features of the individual) as well as a contact network that defines which individuals are related to each other.

The problem the authors describe is interesting and important. It has seen substantial attention in the ML-Healthcare community in recent years. Nevertheless there are some confusing aspects to the paper that make it hard to evaluate the efficacy of the proposed method.

Comments as they appear in order in the paper (comments with higher importance denoted with **):

The authors' use of the variable e for exposure state is a bit confusing and inconsistent. In S3P1 they define e as a sum of contacts' exposures and so it should be an integer. Later in S4P1, they describe a failure mode in which e_a = .9 and e_b = .1, which would suggest the authors are viewing e as on a spectrum from 0 (little exposure) to 1 (a lot of exposure). It's also not clear, throughout, how exposure (and the features x) relates to infection; are there example joint distributions or conditional distributions of y given e that could describe that relationship? Is it being treated merely as another feature (normalized/standardized for the purposes of the NN model) or does it have some special status in how it relates to x^bar and y?

**In S3P2, the authors define biased testing as entailing p(Xt | Ot = 1) \neq P(Xt | Ot = 0) \neq p(Xt). This is rather non-intuitive and it would make more sense for the variables to be flipped: there is bias in testing if the probability of being observed (i.e. tested) is different depending on the patient features. Can the authors clarify why they've chosen to define bias like this?

**In S3P2 the authors state that they assume i's outcome is independent of their contacts (entirely? just the contacts' outcomes?) given xi. Can the authors clarify why this might be a reasonable assumption? The authors, for instance, don't specify any notion of time in making this assumption and so really it seems to be saying y_i^t \indep y_j^t | x_i^1, x_i^2, ..., x_i^t. This is potentially problematic though since it would make sense for y_i^t to depend on y_j^{t-1} (my contact's true infection status yesterday affects my infection status today) and it's obvious that y_j^t and y_j^{t-1} should be dependent. The authors should consider providing a figure (e.g. non-causal DAG) that illustrates these assumptions. As is, I'm very concerned that either the assumption is not sufficient to eliminate network dependency biases, or that the assumption isn't valid in the data the authors are envisioning their method being applied to, or both.

S3P2: J_i^t \in D_0 \cup D_1 should be \subset (or \subseteq) rather than \in.

S3P3: The authors define the inverse probability of being tested as p(Ot = o)/p(Ot = o | Xt). This is also a bit weird of a way to define it: the probability of being tested is the authors' numerator P(Ot = o) and the conditional probability of being tested (or propensity for being tested) is the authors' denominator p(Ot = o | Xt) so it doesn't make sense that the inverse should be this fraction rather than just 1/(the numerator).

**S3P3: Using the authors' definition of the inverse probability of being tested, it's not clear how they obtain the risk in Eq. 1. Starting with the risk in the paragraph above, Rf = E[l(f(Xt), Y^{t+1})], conditioning on O = 1 corresponds to dividing by p(O = 1). To balance this, since the LHS of Eq. 1 is the same risk Rf, it makes sense that the authors multiply the loss by p(Ot = o), the numerator of w. It is not clear how dividing by the denominator of w (thus just multiplying the whole loss by w) results in the same risk. It's unclear to me whether this is a confusion on probability algebra or these some other assumption being made here (overlap should not be sufficient).

S3P4: "However, the following reweighted empirical loss is an unbiased estimator of (...)" -- how do we know it's unbiased? Is there a proof?

S4P1: What is the reason for adding 'hats' (^) to the y's in the sentence "This becomes obvious if we breakdown \hat{e}_i^t draws from Q as follows (...)". Is the idea that we're iteratively plugging in the predictions for the contacts' infection status? This seems to be the first time in the paper this sort of idea is hinted at and so it's not clear what the estimate is meant to be.

**S4.1P1 "We stress that we do not require \hat{y} to be an accuracte estimate of the true labels, but only require that there is significant separation between the imputed values (...)" Is this formalized anywhere? Can the authors please provide equations and math to provide intuition for why this might be the case if not a formal proof? Empirical evidence isn't exactly enough since it's feasible for there to be other explanations for "good" performance.

**S4.1P2 The same comment as above applies to the hand wavy description of the cluster assumption at the bottom of the next paragraph. "The clusers assumtion states that (..) exposure states tend to form near discrete clusters" -- what's meant by 'tend to'? More intution would be nice here.

S5P1 "Let A_i^t be the set of ancestors" -- in what sense? Is there a (potentially causal) graph underlying your model? If not, what is an ancestor?

S5P1 What is \mathcal{U}^t? This doesn't appear to be defined

**General: What assumptions are being made about the missingness mechanism for the true infection states. The authors should be very clear by stating something along the lines of "we're assuming missingness at random". That seems to be what the authors are getting at with their conditional independence assumption but i) it's not obvious that assumption is sufficient (see above) and ii) the authors haven't done a great job of linking that assumption to their argument for the efficacy (identification) of their model.

S5P2/3: The authors describe the method by which they determine convergence of model fit -- I'm curious, given that they cite Chernozukov 2017, what properties their model has wrt sample efficiency and convergence to the "truth" (i.e. the parameters of their model converge to the true parameters of the function that maps X to Y)? In general, I'm not aware of theory that gives convergence rates for neural networks in the way Chernozukov discusses and so I'd be concerned the authors might be picking an arbitrary convergence cutoff rather than something principled.

S6.1P1: "Our goal here is to highlight how MIINT can be used to inform testing and isolation policies that lead to reduction in infection rates" -- the authors should be _very_ careful with using this sort of language. It implies a sort of ground truth causal efficacy to the work when this is not causal work. I agree that semi-supervised approaches are very similar to causal methods but the authors do not appear to spend sufficient time considering the assumptions necessary to ensure valid causal inferences, rather than simply fitting a black-box prediction model.

Simulation study: what is the mechanism for deciding whether somewhat gets tested? It isn't obvious in the description how this selection works. Additionally, from the description, it's not exactly clear what the probability of actually being infected is, given the features (from MNIST) and the contacts (indirectly from MNIST?)

Sensitivity to the potency property: "This confirms out conjecture (...)" The authors should be careful with such strong language based on empirical, simulated evidence.

Minor:
Throughout the paper there are several instances of incorrect grammar or other writing issues that make it hard to determine the meaning of the sentence in question. For instance in the first paragraph of S2, the authors say "In the machine learning literature, previous work has relied on proxies for exposure, e.g., the prevalence of a disease in a community (citation), and implicitly assume that conditioning on individual characteristics." The authors should carefully proofread to remove these artifacts as there are some instances where they pose a more substantive impact on the readability of the paper.

---

> ### Author Response · Authors · 2020-11-25
> **Clarifying confusions (1 of 2)**
>
> We thank the reviewer for the thorough review, and insightful comments. We believe that most of the confusions are related to gaps in our explanations. We will address that by including additional explanations, and citations to other work and the existing theoretical results which our work builds upon.
>
> 1. The definition of exposure is  in section 3 paragraph 1. It is the sum of neighbors’ observed infection states (for neighbors who were tested), or the imputed infection states if the neighbors were not tested. For neighbors who were tested, their contribution to the main patient’s exposure would either be 1 or 0. For neighbors who were not tested, their contribution to the main patient’s exposure would be some value between 0 and 1. In the example that the reviewer mentions, we were considering the case where the main patient had one neighbor who was untested. In that case, the total exposure state is equal to the neighbor’s imputed infection state. Consider another example: if the patient had 4 neighbors, 1 tested positive, 1 tested negative, 1 had an imputed probability of infection = 0.1, and the last neighbor has an imputed probability of 0.8. The main patient’s exposure state, e, would be = 1 + 0 + 0.1 + 0.8 = 1.9. The exposure state is treated as another feature in the input state (by definition of x in section 3 paragraph 1)
>
> 2. In S3P2, the statement about p(Xt | Ot = 1) \neq P(Xt | Ot = 0) \neq p(Xt) is a way of describing biased data, and is commonly used in domain shift, causal inference, and offline reinforcement learning literature (see Cortes 2010, and Cortes 2008 in the main paper). One way to understand it is: the set of patients for whom we wish to make predictions, p(x), is not the same as the one for whom we observed tested patients, p(x | o=1), which means that there is a domain shift between the labelled training data, and the data which will be available at test time, p(x).
>
> 3. In S3P2, the assumption of independence is conditional on the neighbors’ true infection state. In other words, the neighbor only affects the main patient through the neighbor’s true infection state. In other words, if we know the neighbor’s true infection state, we do not need to know anything else about the neighbor in order to estimate the main patient’s infection probability. It is correct that y_j^t, and y_j^{t-1} are dependent. In fact our prediction for both the main patient and the neighbor at time t takes into account the predicted or observed infection states at time t-1, which is why we use an RNN. We have explicitly stated that in section 5, paragraph 1.
>
> 4. S3P3, the definition of the inverse probability of being tested as 1/p(Ot = o | Xt) is a common way of reweighting the data such that it gets closer to the “golden scenario,” a randomized testing setting. This setting is very similar to traditional inverse propensity score weighting used when doing causal inference using biased observational data. However, we multiply each of the weights by the marginal distribution, arriving at p(Ot = o)/ p(Ot = o | Xt) as a measure of stability. These two weighting methods are conventional, and widely used in the causal inference literature to deal with biased testing. We have added appropriate citations (Robbins 1998, and Robbins 2000 cited in the main paper).
>
> 5. S3P3: We stress that the risk in Equation (1) can never be obtained or computed because it is defined with respect to the full unobserved distribution. The generalization error in Equation (1) can at best be bounded above by some term that typically depends on the function complexity and the training error. Well-established theoretical results show that our method of weighting yields an unbiased upper bound on the generalization error bounds (e.g., Cortes 2010, and Cortes 2008 cited in the main paper)
>
> 6. S3P4: Cortes 2010, and Cortes 2008 cited in the main paper also show that this estimator is an unbiased estimator of Equation (1).
>
> 7. S4P1, adding hats: the reviewer is exactly correct, the predicted probability of infection for neighbors is plugged back into the model.
>
> 8. S4.1P1, S4.1P2: The cluster assumption and the margin assumption have been described and studied elsewhere. For example, Audibert 2007 and Rigollet 2007 cited in the main paper show how bounds on the generalization error can be improved under those 2 conditions.
>
> 9. S5P1: Unobserved ancestors, \mathcal{A},  are defined before Equation (4). For time point 1: the unobserved ancestors of individual i are i’s untested neighbors at time 0, call that set A0. For time point 2: these are the unobserved ancestors of individual i are i’s untested neighbors at time 1, union A0, and so forth.
> 10. S5P1, the definition of \mathcal{U} is in section 3 near the end of paragraph 1

---

> > ### Author Response · Authors · 2020-11-25
> > **Clarifying confusions (2 of 2)**
> >
> > 11. General: We do not assume missingness at random. One of the core issues that our approach aims to mitigate is biased testing, which implies missingness is not random. The problem is easier for missingness at random. The conditional independence assumption does not relate to testing. It simply states that the main patient’s state is independent of their neighbor’s characteristics conditional on the neighbor’s true infection state. In other words, if we know the neighbor’s true infection state, we do not need to know anything else about them to estimate the main patient’s infection state.
> >
> > 12. S5P2/3: Previous results (e.g., in Chernozukov et al) that pertain to nonparametric models (nonparametric in the frequentist not Bayesian sense) are generally applicable to neural networks. Generalization error bounds for our model can be constructed using the same tools that have been presented elsewhere, e.g., section D in Foster et al (citation [1] below, not cited in the main paper). In addition, the convergence rates discussed in the Chernozukov paper, as well as Foster et al are statistical rather than computational convergence rates. So convergence cutoffs are not directly relevant here.
> >
> > 13. S6.1P1. We appreciate the reviewer’s admonition to be cautious with our language. We stress however, that the semi-supervised approach is not the main component in our approach that addresses issues of bias. But rather the inverse weighting scheme that mirrors, almost exactly, the approaches used in causal inference to adjust for biased treatment assignment when learning from observational data.
> >
> > 14. Simulation study: As described in section 6.1 paragraph 5: each model produces a predicted probability of infection for all individuals in the community. The top x% of the patients with highest predicted risk are tested. We sweep over this x value, which is the x-axis in figure 1.
> >
> > 15. Minor: we thank the reviewer for catching that typo, the sentence should have read
> > “In the machine learning literature, previous work has relied on proxies for exposure, e.g., the prevalence of a disease in a community (citation), and implicitly assume that conditioning on individual characteristics along with the exposure proxy is sufficient for prediction”. We have updated the manuscript accordingly.
> >
> > [1] Dylan Foster, Vasilis Syrgkanis, “Orthogonal Statistical Learning” https://arxiv.org/pdf/1901.09036.pdf

---

### Official Review · AnonReviewer1 · 2020-10-29
**Predicting infection on structured data with missing testing**

**Rating:** 5
**Confidence:** 2

**Review:**

This paper formulates the contagious disease into a missing label problem with dependence between each data point. The paper targets an important problem, especially in this pandemic, and the effort is greatly appreciated. However, the writing of this paper is confusing and it makes it hard to catch the main contribution of this paper. There are some concerns:

1. The formulated problem sounds like a node label missing problem in a graph, where the node is patient (x, y) and the edge is whether they are contacted (e). In so, the paper is actually predicting the label of each node. If my understanding is correct, I am not sure why the authors choose the current formulation rather than graph one.

2. The notation and wording are sometimes confusing, especially when I only have limited knowledge in healthcare. For example, even after reading section 4.1, it is still confusing what kind of special data structure the author is indicating. I recommend the author give a more intuitive or even graphic explanation in the paper.

3. All the experiments are self-compared and make the result less convincing. For example, it is not clear whether different NN network would result in different performance because only NN is fixed here.

---

> ### Author Response · Authors · 2020-11-25
> **Relation to graph-based approaches and additional baselines**
>
> We thank the reviewer for recognizing the importance of our work.
>
> 1. Our approach is a graph based approach. Specifically, by considering the neighbors’ contacts, we do take into account the individuals’ neighbors observed states for neighbors who are tested, and the predicted neighbors’ states if they are untested. In addition, we have updated our paper to include additional graph-based benchmarks (GraphSAGE). We’ve also updated the related work section to highlight the similarities and distinctions from other graph based approaches.
>
> 2. The data structure here is the graph or the contact network. A simplification of our approach can be understood as follows: MIINT imputes the exposure state of each individual. This exposure state must be a function of the individuals’ exposure states. MIINT uses that information to regularize out models where the imputed exposure state diverges significantly from the predicted or observed neighbor states. E.g., if one model candidate predicts that individual i is highly exposed, but at the same time predicts that all of i’s neighbors are not infected, this model is heavily penalized.
>
> 3. Our original paper included baselines other than our suggested approach. For example, one of the baselines we included was an oracle model, which has access to the true infection states of each individual at training time, making it a strong, and tough to beat baseline. In addition, we’ve included 2 more baseline (GraphSAGE and Pseudo-labeling). MIINT consistently outperforms both of them. In addition, our neural network approach outperforms ELR, which is the state-of-the art logistic regression in the medical task.

---

### Official Review · AnonReviewer2 · 2020-10-29
**New model and strong simulation framework, somewhat weak baselines**

**Rating:** 7
**Confidence:** 2

**Review:**

In this work, the authors propose an approach, MIINT, for identifying infected individuals using a network-based approach. They also suggest two key properties, potency and similarity among groups, which impact the efficacy of MIINT and similar approaches. A detailed simulation framework is used to compare MIINT to relatively weak baselines. The simulation results show that the MIINT modestly outperforms the baselines; the results also confirm that all approaches degrade as expected as potency decreases and the similarity among groups increases. Experimental results on a (private) real-world dataset are somewhat mixed, but show that MIINT achieves a better true positive rate at an acceptable false positive rate.

The proposed approach formalizes an intuitive understanding of the spread of an infectious disease in (as far as I am aware) a novel way. The authors also identify and empirically evaluate the conditions under which the model breaks down. I also appreciate the effort of creating a simulation model to reflect how the spread may develop over time; I believe the simulation framework could also be useful for other researchers in this area. The model may also be relevant for other network spreader domains, such as “influencers in social media” studies.

My main concerns about the work are about the experiments. First, some sort of graph-based model should be included, such as label propagation or locally-linear embedding (followed by some classifier); of course, standard graph neural network approaches could also be included in a different, more empirical evaluation-oriented study. With the current setup, it is not clear if MIINT in particular, or any approach which accounts for the graph, would outperform the baselines.

Second, it is not clear to me of the practical significance of the “isolation policy” results. Even using an oracle, Figure 1 seems to suggest that between 30-40% of the population would need to be tested to see a reduction in the infection rate. Is that actually reasonable, even in the “closed” clinical environment recommended by the authors in the conclusion?

Minor comments
----------------

The references are not consistently formatted.

Neither the paper nor the supplement describe the features available in the real data.

It seems a bit strange that the AUROC and AUPRC in Table 1 are almost perfectly negatively correlated. Is there any explanation for that?

While I found the text descriptions and intuitions very helpful, all of the in-line equations made parsing many of the paragraphs in Sections 3, 4, and 5 difficult.

“outcomes, we refer” -> “outcomes; we refer”
“If exposed individuals” -> “If exposed, individuals”

---

> ### Author Response · Authors · 2020-11-25
> **Additional baselines, explaining AUPR, and isolation experiment takeaway**
>
> We thank the reviewer for recognizing the relevance of our work to domains even outside of infectious diseases, and for recognizing the novelty of our simulation approach.
>
> 1. We have added 2 new baselines, one is a graph based model (GraphSAGE) and another is a “classical” semi-supervised learning approach (Psuedo labeling) to all simulations and real data experiments. Our original conclusions hold even with the new baselines.
>
> 2. The isolation policy experiment is meant to show our model can reduce the infection rates when it is used to guide testing policies. The magnitude of reduction in infection rate will depend on the characteristics of the pathogen itself (e.g., how contagious it is). In a simulation setting, how contagious a disease is depends on simulation “knobs” that we set ourselves. For example, in our simulation, we consider a highly contagious disease making it necessary to have a non-trivial testing budget to achieve reductions. For less contagious diseases, it might be possible to achieve larger reductions in base infection rates.
>
> 3. We added a detailed description of the features included in the data in the supplement, section C, paragraph starting with “Patient Features”.
>
> 4. As noted in the general comment: we found that models which achieve higher AUPR do so because they have high precision at low recall, whereas our model has the highest precision at high recall values.  High precision at low recall is not necessarily useful in the infectious disease setting, where having a low recall (meaning being unable to identify the true infections) can be disastrous. We added the Precision recall plot in the supplement.

---

### Author Response · Authors · 2020-11-25
**Additional baselines, and explaining AUPRC**

We thank the reviewers for recognizing the relevance, and importance of our work. Most importantly, the reviewers urged us to implement benchmarks that fall into two categories: semi-supervised methods, and graph based methods. We have implemented and reported the results for those two methods, and updated the manuscript showing that our previous conclusions hold: our approach does better than the two new benchmarks. The reviewers’ feedback helped us realize an additional strength of our approach: it combines the strengths of semi-supervised learning methods (e.g., label propagation methods) which make use of unlabeled data, and graph based methods (e.g., GraphSAGE), which makes use of rich relational data. We thank the reviewers for guiding us to realize this interesting connection, and  have updated the related work to reflect this property.

In addition, several of the reviewers raised the issue that while our model outperforms all benchmarks on AUROC, and TPR at FPR = 10%, it underperforms compared to some of the benchmarks for the AUPR. One of those benchmarks (POM) is a partial oracle model that has access to results from PCR test, which our model and other benchmarks do not have access to. In that it should come as no surprise that POM has the highest AUPR. In addition, we found that models which achieve a higher AUPR than do so because they have high precision at low recall, whereas our model has the highest precision at high recall values.  High precision at low recall is not necessarily useful in the infectious disease setting, where having a low recall (meaning being unable to identify the true infections) can be disastrous. We added the Precision recall plot in the supplement.

---

### Decision · Program_Chairs · 2021-01-07
**Final Decision**

**Decision:**

Reject

**Comment:**

This paper studies a timely problem and consider an interesting approach, but overall there were many concerns about technical details and the validity of the framework. The positive reviewer also mentioned concerns about the experiments, which others also found to be an insular comparison with weak baselines. Following the response period, in discussion there are additional concerns arising related to the lack of details, for instance related to possible unidentifiability of the model. As one reviewer discusses,  the authors are attempting to use RNNs to impute missing infection status labels when the missingness mechanism is assumed to be (i) not at random, (ii) playing out over time (as it is unclear whether Y^t is assumed (conditionally) independent of Y^t' with t' << t), and (iii) subject to interference (whether someone is tested is the 'treatment' here since it's a missingess problem and one person's propensity to be tested could causally affect another person's downstream infection status since apparently no Markov independence is assumed. There is also consensus that the writing quality can be greatly improved. Overall this work contains some ideas with potential in a thorough revision